# Super-resolution orbital angular momentum holography

Zijian Shi[1,2,3], Zhensong Wan[1,2,3], Ziyu Zhan[1,2,3], Kaige Liu[1,2,3], Qiang Liu[1,2,3] ✉ & Xing Fu [1,2,3] ✉

Computer-generated holograms are crucial for a wide range of applications such as 3D displays, information encryption, data storage, and opto-electronic computing. Orbital angular momentum (OAM), as a new degree of freedom with infinite orthogonal states, has been employed to expand the hologram bandwidth. However, in order to reduce strong multiplexing crosstalk, OAM holography suffers from a fundamental sampling criterion that the image sampling distance should be no less than the diameter of largest addressable OAM mode, which severely hinders the increase in resolution and capacity. Here we establish a comprehensive model on multiplexing crosstalk in OAM holography, propose a pseudo incoherent approach that is almost crosstalk-free, and demonstrate an analogous coherent solution by temporal multiplexing, which dramatically eliminates the crosstalk and largely relaxes the constraint upon sampling condition of OAM holography, exhibiting a remarkable resolution enhancement by several times, far beyond the conventional resolution limit of OAM holography, as well as a large scaling of OAM multiplexing capacity at fixed resolution. Our method enables OAM-multiplexed holographic reconstruction with high quality, high resolution, and high capacity, offering an efficient and practical route towards the future high-performance holographic systems.

Computer-generated holograms (CGHs) have received increasing attention in recent years because of its flexibility in manipulating optical wavefronts. Nowadays CGHs have been used in many applications, such as optoelectronic computing[1,2], three-dimensional displays[3–9], beam shaping[10–12], and optical encryption[13–15]. Holographic multiplexing is an efficient solution to overcome the difficulty of generating CGHs in real-time, employing multiple physical degrees of freedom (DoFs) such as polarization[16–18], wavelength, and incident angles[19–21], but these DoFs have quite limited bandwidth. As a new DoF of multiplexing, the orbital angular momentum (OAM) of an optical vortex beam is very promising because of its theoretically unbounded orthogonal modes[22–25]. Recently, OAM holography was proposed, which offers the possibility to improve the bandwidth of multiplexed

holography, and has been implemented on various platforms, including phase-only spatial light modulator (SLM)[26], phase-only metasurface[27], and complex-amplitude metasurface[28].

In OAM holography, each pixel of the image retains the properties of the incident OAM beam by discretely sampling the target image in the Fourier plane of the holographic system; thus, a set of OAMs of incident beams can be employed as information carriers[26], which however inevitably causes the resolution degradation with respect to the original image (Fig. 1a). Specifically, the ratio of the diameter of largest addressable OAM mode ($d_{max}$, denoting the OAM-multiplexing capacity) to the sampling distance ($L$, indicating the image resolution of OAM holography), $\gamma = d_{max}/L$, is a core factor termed as the sampling condition. It is greatly desirable to enlarge the value of $\gamma$ as much as

[1]Department of Precision Instrument, Tsinghua University, Beijing 100084, China. [2]State Key Laboratory of Precision Space-time Information Sensing Technology, Beijing 100084, China. [3]Key Laboratory of Photonic Control Technology (Tsinghua University), Ministry of Education, Beijing 100084, China. ✉e-mail: qiangliu@tsinghua.edu.cn; fuxing@tsinghua.edu.cn

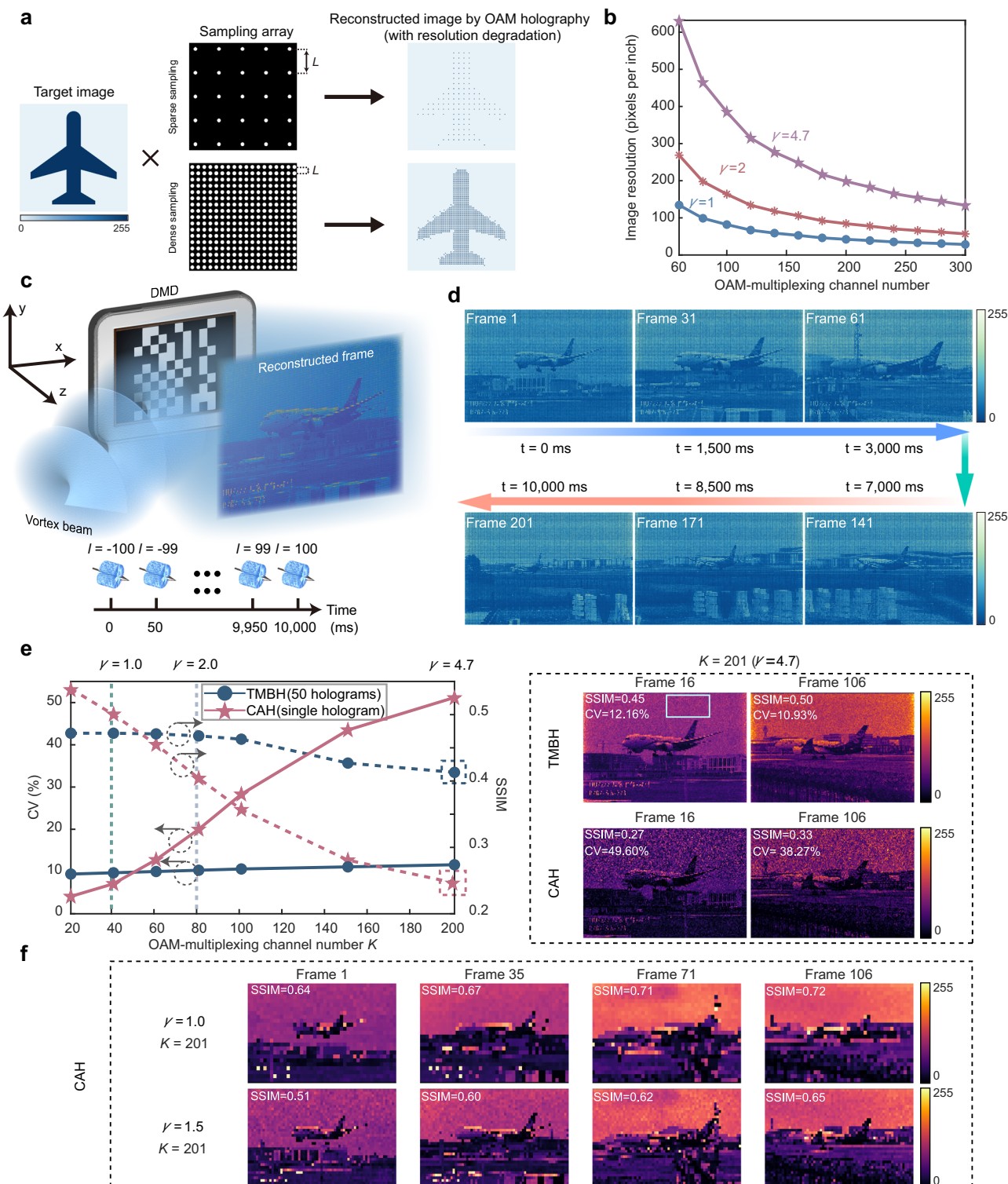

**Fig. 1 | Holographic video display beyond the resolution limit by temporal multiplexing binary OAM holography (TMBH). a** Schematic illustration of varied degrees of resolution degradation by sparse and dense discrete samplings in OAM holography. **b** Advantage of enlarging the sampling condition $\gamma$ beyond unity in obtaining higher resolution (super-resolution) and higher capacity of OAM holography. **c** Schematic illustration of TMBH setup based on DMD. **d** Reconstructed super-resolution frames of the holographic video by the TMBH method, with sampling condition of $\gamma = 4.7$ and multiplexing channel number of $K = 201$. **e** Evolution of reconstructed image quality of TMBH and CAH methods based on the averaging results of initial 21 frames, evaluated by the metrics of SSIM and CV, versus the growing multiplexing channels. The empty box in the reconstructed frame marks the area where the CV is calculated for all frames. **f** Reconstructed frames by the CAH method at $\gamma = 1$ (corresponding to the conventional resolution limit at 201 multiplexing channels) and $\gamma = 1.5$.

possible for OAM holography, since it corresponds to higher resolution with a certain number of OAM channels, or equivalently higher capacity at a certain resolution (Fig. 1b, see also Supplementary Note 1 for the calculation method of resolution).

However, for the general coherent OAM holography as previously reported, OAM property cannot be preserved in densely sampled high-resolution images due to interference, thereby imposing a fundamental resolution limit for a given OAM channel number by demanding $\gamma \leq 1$[26–28], which severely suppresses the potentials of OAM holographic multiplexing towards high resolution and high capacity. Moreover, even under the operating condition of $\gamma \leq 1$, OAM-multiplexing holography is not immune to multiplexing crosstalk during the image reconstruction, and in particular, OAM beams with a small helical mode index interval ($\Delta l$ ~1) suffer from strong multiplexing crosstalk[29]. Due to these main bottlenecks, previous demonstrations of OAM holography usually set $\Delta l$ above 2, sacrificing the multiplexing capacity, and the images reconstructed in experiments were mostly limited to low-resolution ones such as simple icons and basic geometric shapes[26–28,30–33]. Even though a recent approach of complex-amplitude OAM holography (CAH) based on metasurface[28] reduced the multiplexing crosstalk to some degree, it was still clamped by $\gamma \leq 1$.

In this work, we break the resolution barrier ($L = d_{max}$) of OAM holography imposed by the sampling criterion of $\gamma \leq 1$, for the first time to the best of our knowledge. We realize high-quality reconstruction of OAM holography at $\gamma$ far beyond 1, which signifies a much smaller sampling distance $L < d_{max}$ compared with the previous approach using the same set of OAM-multiplexing channels (same $d_{max}$), i.e., achieving the super-resolution OAM holography. Equivalently, OAM-multiplexing capacity can be remarkably boosted (larger $d_{max}$) while maintaining the resolution (same $L$). It should be emphasized that the super-resolution concept in this work applies only to OAM holography that the case of $\gamma = 1$ represents the OAM holography at the resolution limit, and the cases of $\gamma > 1$ represent the super-resolution OAM holography. Comparison between the resolution limit in OAM holography and that in a conventional optical system is illustrated in Supplementary Note 1.

Specifically, we establish a comprehensive model on multiplexing crosstalk from two types of interference in OAM holography, and propose a pseudo incoherent case that is almost crosstalk-free for varying $\gamma$. Then we approximate the pseudo incoherent case in an alternative but realistic approach, by temporal multiplexing of coherent light, dramatically eliminating the multiplexing crosstalk and successfully obtaining high-quality super-resolution performance (Fig. 1d, e), where the image quality is assessed by the metrics of coefficient of variation (CV, see Methods) and structural similarity index measure (SSIM, see Methods). The temporal multiplexing binary OAM holography (TMBH) proposed in this work enables the OAM-addressed 201-frame video display (with a grayscale of 256) at a super-resolution of 200 × 200 pixels in one frame (see Fig. 1 and Supplementary Movie 1), largely enhancing the reconstructed resolution by a factor of 4.7, compared with the resolution limit of previous state-of-the-art coherent OAM holography[26–28]. In addition, we demonstrate the high-quality super-resolution reconstruction of binary images, with the resolution enhancement by a factor of 9.2. The approach proposed enables OAM-multiplexed holographic reconstruction with high quality, high resolution, and high capacity.

## Results
### Spatial frequency distributions in coherent and incoherent OAM holography

For the general coherent case (Fig. 2a) that uses a coherent illumination source, the constituent spatial frequencies ($k_g$ in the momentum space) of an OAM-conserving hologram add a linear spatial frequency shift to an incident OAM beam ($k_{in}$)[26,27]. Considering a hologram that multiplexes two OAM channels as an example, a sparse sampling ($\gamma = 0.56$) can retain the OAM property of the incident beam (as denoted in the enlarged area $i$), however a denser sampling with $\gamma = 1.75$, which exceeds the sampling criterion limit, causes a complete loss of OAM property due to interference (see the enlarged area $j$), the reason of which will be studied in next subsection. On the other hand, for the general incoherent case (Fig. 2b) that uses an incoherent illumination source, the spatial frequencies of different incoherent wavefronts have deviated outgoing directions according to the Bragg

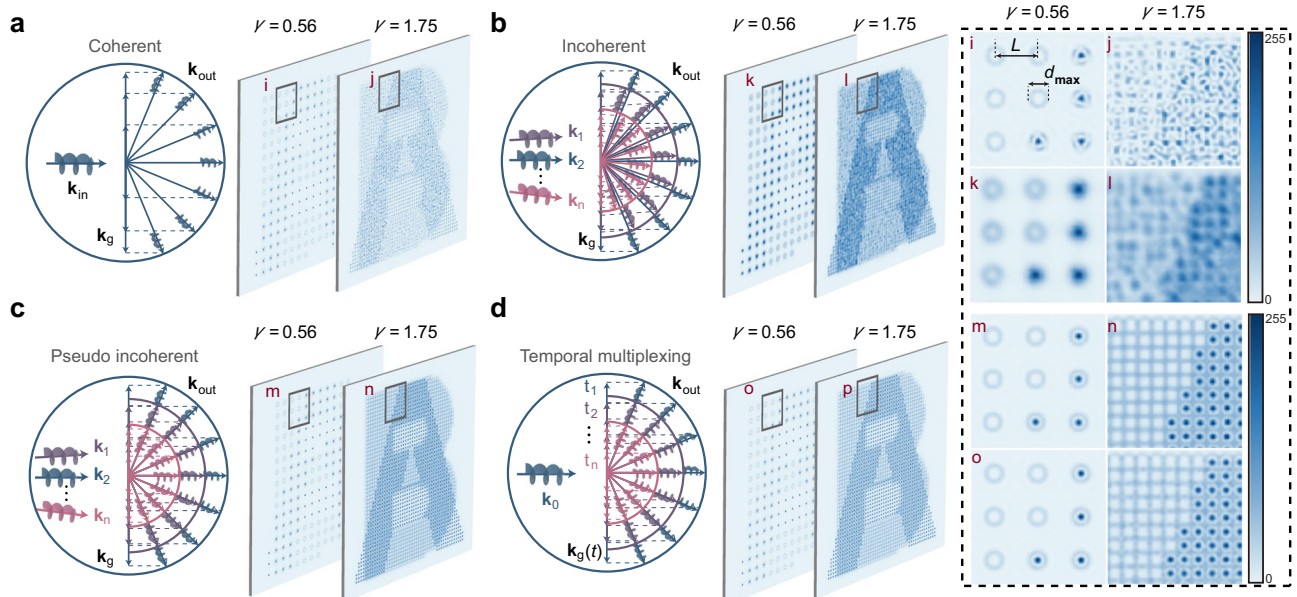

**Fig. 2 | Schematic illustration of the OAM property transfer in the spatial frequency domain. a–d** Reconstructed images with two OAM channels are shown, where the enlarged areas (marked as $i$ to $p$) are compared in the rightmost panel.

**a** General coherent case. **b** General incoherent case. **c** Pseudo incoherent case. **d** Coherence suppression case with temporal multiplexing.

diffraction formula, which brings serious undesired blurring to the reconstructed image (see the enlarged area of *l*).

Here we purpose a pseudo incoherent case (Fig. 2c), where, unlike the general incoherent case, we assume the outgoing wave vectors that leave the hologram coincide perfectly for all incoherent incident vectors ($k_1$ to $k_n$), so that individual diffraction patterns can be superposed in an intensity manner (see the enlarged area of *n*), without introducing blurring at all. However, this case is practically challenging to realize due to the violation of the Bragg diffraction formula; thus, a coherent analogy to that is alternatively carried out in this work, by temporal multiplexing of a coherent beam, thereby suppressing the coherence. It is shown in Fig. 2d that temporal multiplexing introduces time-varying phase delays to the incident vector, thereby creating multiple incoherent wavefronts within each modulating duration (see also Supplementary Note 2), which is akin to inherent multiple wavefronts in the pseudo incoherent case, and consequently enables high reconstruction quality at super-resolution condition as well (see the enlarged area of *p*).

## Two types of interference in OAM holography

The reconstructed image of OAM holography is composed of a series of discrete pixels, and each pixel location contains one or more OAM modes. Before examining the multiplexing crosstalk and sampling criterion limit of coherent OAM holography, we present that there exist two types of interference for each pixel of the reconstructed image: (1) interference between different OAM channels at the same pixel (see Supplementary Note 3), termed as same pixel interference (SPI); (2) interference between OAM modes in adjacent pixels, termed as adjacent pixel interference (API), referring to the effect that OAM modes at adjacent pixels have interference and destroy the structure of a reconstructed image, when the distance between the discrete pixels is smaller than the diameter of OAM modes. We clarify in the following that SPI mainly arises from a small index interval (especially when $\Delta l = 1$), while the degree of API is influenced by both $\gamma$ and $l_{max}$, the latter being the largest addressable OAM index.

Firstly, we consider the scenario of $\gamma < <1$, in which API is negligible since the OAM channels from adjacent pixels in the reconstructed image do not overlap with each other; hence only SPI may take effect. SPI is manifested by an instance shown in Fig. 3, where five sparsely sampled images of alphabet letters are encoded into a complex-amplitude OAM-multiplexed hologram (with helical mode indexes of $l = 0, -1, -2, -3, -4$) and the target images have a sampling interval of $L = 30 \lambda/NA$, corresponding to $\gamma = 0.4$, where $\lambda$ and NA are the wavelength and numerical aperture of the holographic system respectively (see also Supplementary Note 1). When an OAM beam with $l = 1$ is incident on the hologram, five images carried by different OAM channels are displayed simultaneously on the image plane (Fig. 3a). For the coherent illumination case, these images are coherently superposed (top right corner of Fig. 3a); thus OAM mode in each pixel is replaced by various intensity patterns, due to interference between OAM modes from different channels with random phases. In particular, the intensity profiles of three representative superposed mode pixels (pixel q, r, and s) in the reconstructed image are shown in the left panel of Fig. 3b (experimentally observed superposed mode pixels are shown in Supplementary Fig. 21a), along with cross-sectional views. Note that the two vertical dotted lines in the subfigure of Fig. 3b indicate a circle area with full width at 70% of the maximum fundamental mode ($l = 0$) intensity profile, and this area is defined as the intensity integral region, hereinafter in this work, for calculating the intensity of a pixel in all cases. One can see the intensity peaks of pixels q, r, s are off-centered due to SPI, resulting in largely varied intensity values (ranging from 58% to 100%, normalized to the intensity of fundamental mode) and thus introducing undesired intensity fluctuation in the post-processed decoded image (Fig. 3c, dashed line).

In contrast, for the pseudo incoherent case, the OAM modes with centrosymmetric intensity distribution is well maintained since all pixels are superposed in an intensity manner. As such, the Gaussian fundamental modes (at signal locations) and the OAM modes (at non-signal locations) are distinctively shown in the reconstruction results (bottom right corner of Fig. 3a, experimentally observed Gaussian fundamental modes and the OAM modes are shown in Supplementary Fig. 21b). The cross-sectional views in the left panel of Fig. 3b (pixel u, v, and w) indicates no center offset, and the normalized intensities among different pixels are much more consistent, ranging from 91 to 97%. Consequently, post-processed decoded images in the pseudo incoherent case exhibit much lower intensity fluctuation (Fig. 3c, solid line) and almost crosstalk-free feature. Notably, SPI becomes weaker as $\Delta l$ increases, due to a much smaller spatial overlap among different OAM channels at the same pixel (see Supplementary Note 4).

Secondly, we consider the scenario of $\gamma > 1$, where the OAM channels from adjacent pixels cannot be completely separated from each other, thus, API starts to take effect and may introduce heavy crosstalk in the reconstructed image. To independently investigate the influence of API on the signal location of the reconstructed image, we examine an OAM-multiplexed example of all four OAM modes ($l = 0, 5, 10, 15$) as shown in Fig. 4a, where $\Delta l$ is so large as 5 that SPI can be ignored (example with $\Delta l = 1$ is shown in Supplementary Note 5), and $d_{max}$, the diameter of OAM mode with $l = 15$ is marked. Specifically, we study the pixel of interest (marked as 0) at a signal location and its eight adjacent pixels (marked from 1 to 8), all incorporating four OAM modes as stated. Note that those eight adjacent pixels are assumed to have the same intensity here, while a more general API case considering the spot array with different intensities is studied in Supplementary Note 6. The API patterns in the coherent case are illustrated in the top row of Fig. 4b, compared with the intensity superposition pattern between adjacent pixels in the pseudo incoherent case. The bottom row of Fig. 4b describes the cross-sectional views of the intensity profile, implying that for the coherent case, API makes the intensity distribution at pixel 0 changes significantly from the peak value of 1.0 (red dotted line) to 1.5 (solid line), while it remains almost unchanged for the pseudo incoherent case.

One can tell from the top row of Fig. 4c that for the coherent case, the intensity fluctuation, as evaluated by the CV, rises rapidly as $\gamma$ enlarges from 4% at $\gamma = 1$ to 36% at $\gamma = 2.9$, which is also evident by the intensity values on 100 trials at the bottom row of Fig. 4c. As such, in order to suppress undesired intensity fluctuation, the upper limit of the multiplexing channel number in the coherent OAM holography is strictly set at $\gamma = 1$, where the outermost modes of adjacent pixels just intersect (see the upper left inset of Fig. 4c). In contrast, the pseudo incoherent case ultimately breaks this limit, allowing the reconstruction of signal location with zero CV and absolutely no intensity fluctuation at the condition far beyond $\gamma = 1$ (equivalently to supporting much more OAM channels at the same resolution).

In addition, the schematic of Fig. 4d (4e) is similar to Fig. 4b (4c), except that each pixel contains only three OAM modes ($l = 5, 10, 15$), in order to illustrate the influence of API on a non-signal location. Figure 4e shows that the average intensity at the non-signal location of the reconstructed image increases sharply when $\gamma > 2$ for both coherent and incoherent case, but remains at a rather low level relative to the signal intensity in Fig. 4c, because the outermost mode with $l_{max}$ as high as 15 has very weak intensity and barely contributes to adjacent pixels.

In short, the key to achieving high-resolution OAM holography is to attenuate the intensity fluctuations of the reconstructed image, while the background noise, which represents a constant image offset, has little effect on the quality of the reconstructed image when the multiplexed number is relatively small. The models of SPI and API for the case of temporal multiplexing of 50 holograms are presented

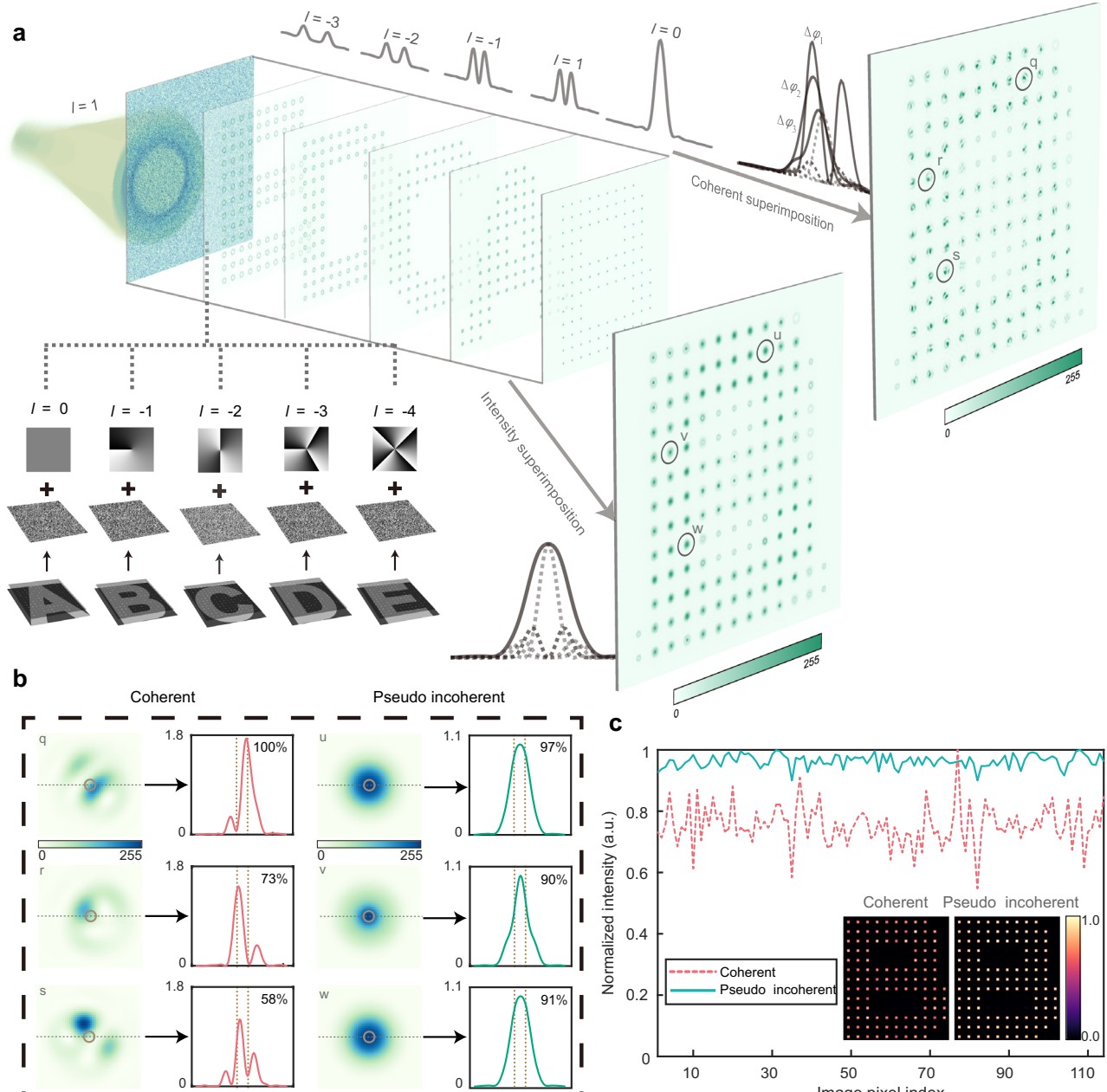

**Fig. 3 | Two types of interference in OAM reconstructed images. a** OAM beams with the helical mode index $l = 1$ is incident on the OAM-multiplexed hologram, and images carried by different OAM channels are displayed simultaneously (left). In the coherent case (top right), these coherently superposed images have the interference of different OAM modes at all pixels. In the pseudo incoherent case (bottom right), these images are superposed by the intensity and demonstrate no interference. **b** Three representative pixels from the reconstructed image in **a**, for the coherent (left) and pseudo incoherent (right) cases. Intensity distributions are presented in both 2D profiles and 1D cross-sectional views where the values indicate the normalized intensity integral within the region as denoted by vertical dotted lines. **c** Intensities of all pixels in the post-processed results of the reconstructed image in **a**.

in Supplementary Note 7. The combinative effect of SPI and API is illustrated in Supplementary Note 8.

**Principle of temporal multiplexing binary OAM holography**
The coherent analogy to the pseudo incoherent case is inspired by temporal multiplexing holography[34–36], where the reconstructed images are constantly updated by refreshing CGHs with the same image but uncorrelated initial random phases at a high frame rate, leading to the reconstructed images observed being the accumulation of multiple independent images in a narrow time duration, thereby dramatically reducing the degree of coherence in the reconstructed results. This method requires high-speed spatial light modulating devices such

as digital micromirror device (DMD) and ferro-electronic liquid crystal on silicon (FLCOS), thus restricting the available types of CGHs to amplitude binary holograms (for DMD) and phase binary holograms (for FLCOS). Here we use amplitude binary holograms for simulation and experiment.

The design process of a binary OAM-selective hologram is shown in Fig. 5a. To achieve the OAM holography, a sampling array with a random phase function samples the target image in the image plane[28]. Then the inverse Fourier transform of the sampled image yields a complex-amplitude OAM hologram. To achieve OAM selectivity, the phase portion of the complex-amplitude OAM hologram is added with a helical phase mask. We encode the complex-amplitude

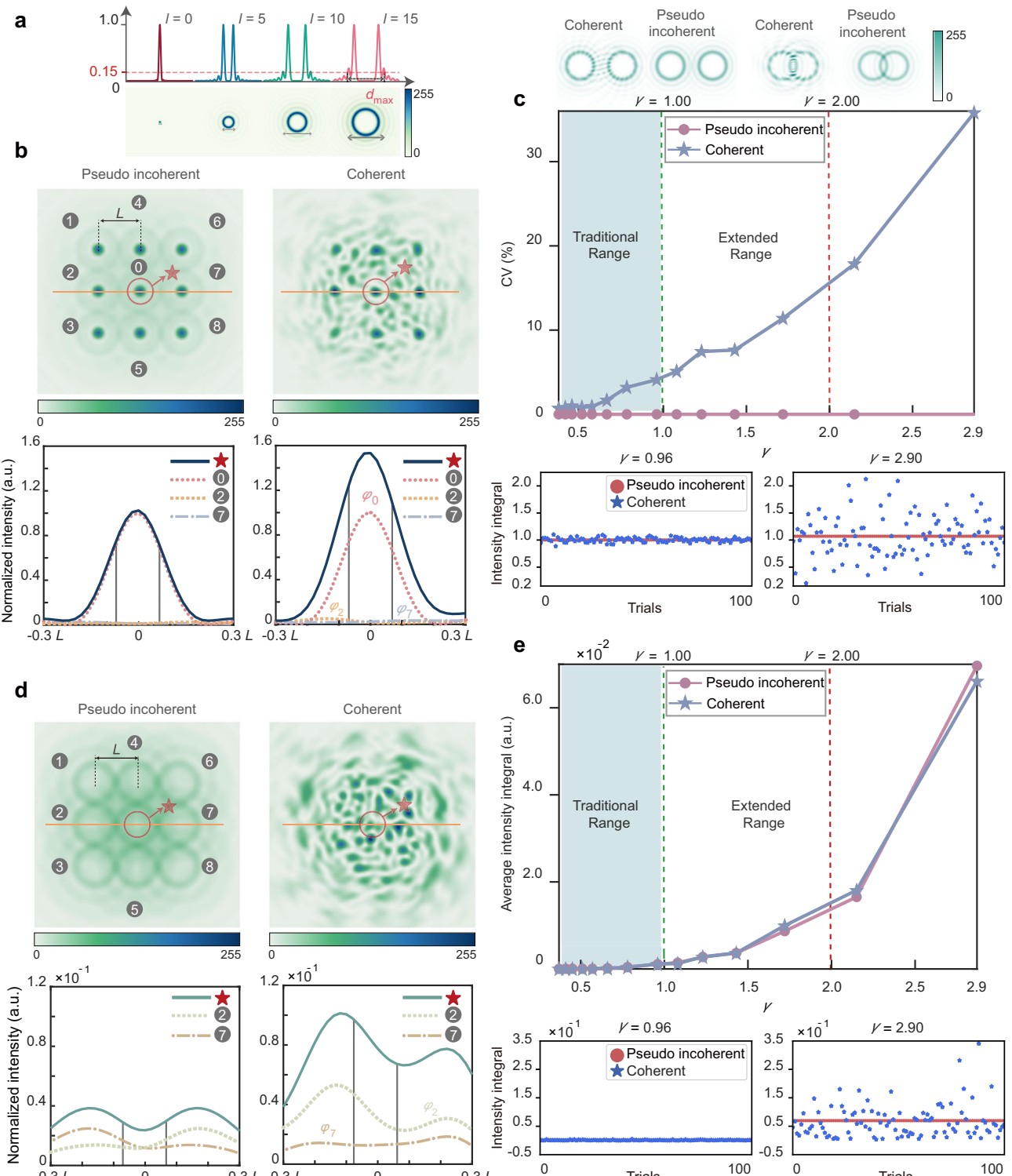

**Fig. 4 | General interference model between adjacent pixels. a** OAM modes involved in the multiplexing. **b** Schematic of API effect on a signal location, with densely packed OAM modes ($l = 0, 5, 10, 15$) at nine adjacent locations (top) as well as cross-sectional views (bottom). **c** Fluctuations of intensity integral in **b** versus γ. The CV value indicating the intensity fluctuations is calculated from 100 trials. Specific intensity results of each trial are shown for the cases of $γ = 0.96$ and $γ = 2.90$ (bottom). **d** Schematic of API effect on a non-signal location, with densely packed OAM modes ($l = 5, 10, 15$) at nine adjacent locations (top) as well as cross-sectional views (bottom). **e** Average intensity integral in **d** versus γ.

OAM-selective hologram into an amplitude OAM-selective hologram using a modified off-axis encoding method[37], then binarize the amplitude OAM-selective hologram to obtain a binary OAM-selective hologram. To suppress the inherent binary quantization noise from the binary holograms, here we use the SWMAE algorithm[38]

(see Supplementary Note 9) to create a low-noise region in the spatial frequency domain and successfully achieve binary OAM holograms experimentally (see Supplementary Fig. 22).

The flowchart of the TMBH method is illustrated in Fig. 5b. First, we select multiple target images to be encoded. Each image is

**Fig. 5 | Design principles of binary OAM-selective holograms and OAM-multiplexed holograms for temporal multiplexing. a** Design approach for a binary OAM-selective hologram. **b** Flowchart of generating binary OAM-multiplexed holograms.

transformed into a complex-amplitude OAM hologram $\Omega$ according to Fig. 5a. Next, in each channel a specific helical phase mask is added to the phase of the complex-amplitude OAM hologram, in order to create multiple orthogonal, independent complex-amplitude OAM-selective holograms. Then these OAM-selective holograms are superposed in a complex-amplitude manner to synthesize a complex-amplitude OAM-multiplexed hologram. Finally, after amplitude encoding and binary quantization, a binary OAM-multiplexed hologram $N$ is obtained. A series of binary OAM-multiplexed holograms are generated by repeating the above process. In different loops, different random phase functions $\Psi$ are added to each image, which results in different arrangements of binary OAM holograms with the same multiplexed information. In order to illustrate the similarity of TMBH with the pseudo incoherent

case in detail, the temporal multiplexing results of 50 binary holograms are shown in Supplementary Note 10. The analysis of OAM content in the reconstructed image of TMBH and that of a single moment is presented in Supplementary Note 11.

## Characterization of temporal multiplexing binary OAM holography

In this subsection, we compare in detail the performances of three classes of OAM holograms: phase-only OAM holography (PH, with single hologram), CAH (with single hologram), and TMBH (with 50 holograms temporally multiplexed) proposed in this work. For the first example of illustration, a set of $300 \times 300$ binary images of winter sport icons are encoded separately into multiple OAM channels, with the sampling distance of $L = 8 \, \lambda/\text{NA}$ and index interval of $\Delta l = 1$. We

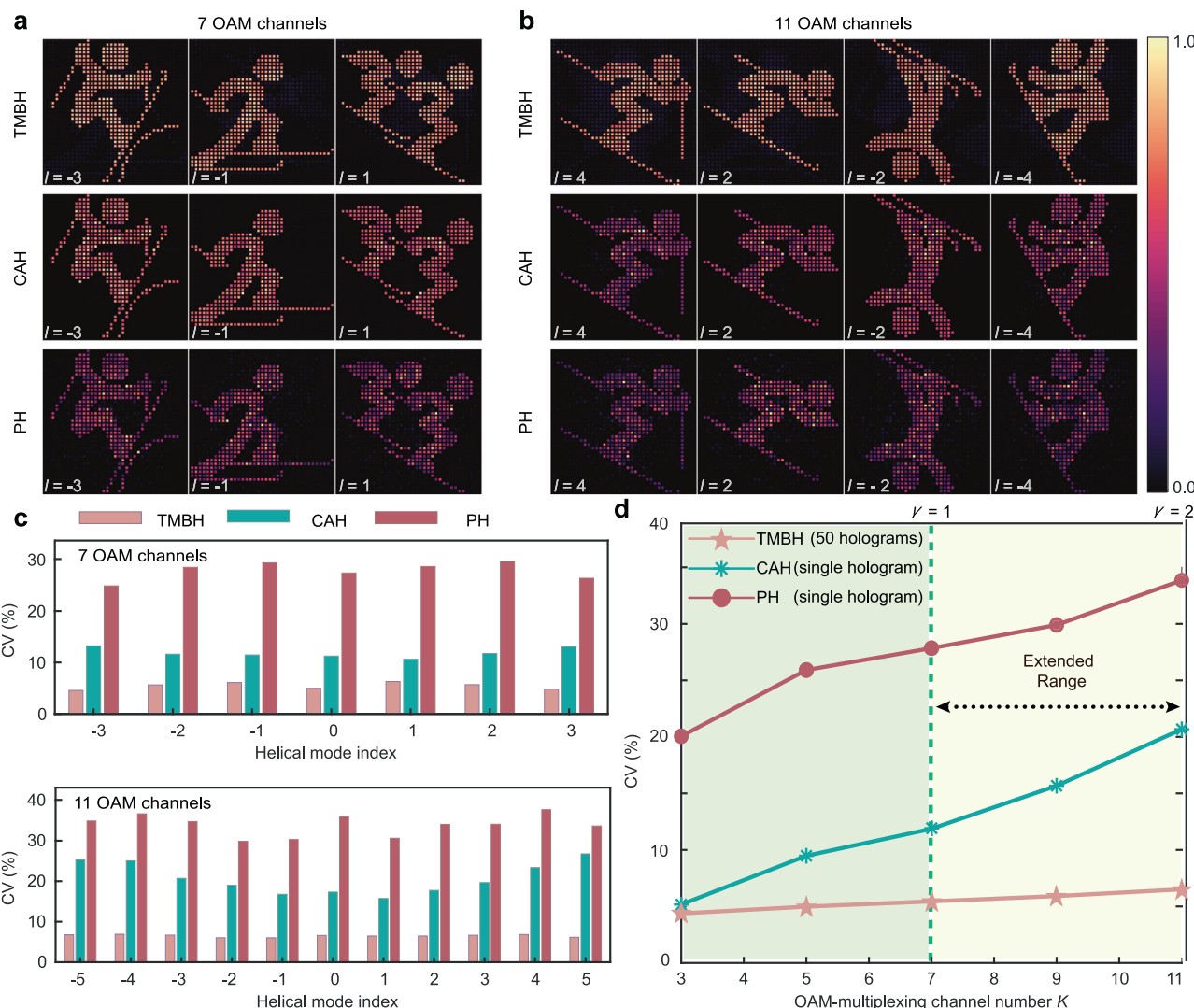

**Fig. 6 | Numerical reconstruction results of binary images by three classes of OAM holography. a** Reconstructed results (partially shown) with 7 OAM channels. **b** Reconstructed results (partially shown) with 11 OAM channels. **c** Quantitative comparison of reconstruction quality. **d** Evolution of intensity fluctuation of reconstructed images versus the number of OAM-multiplexing channels, where the multiplexing channel numbers for $\gamma = 1$ and $\gamma = 2$ are marked.

numerically generate three classes of OAM-multiplexed holograms to carry information from multiple OAM channels. The reconstructed results for 7 and 11 OAM channels are demonstrated in Figs. 6a, b respectively, indicating that the TMBH method apparently has the best image quality and lowest CV value in each channel (see also Fig. 6c). Figure 6d shows TMBH has the CV value of 5.5% at the traditional boundary of $\gamma = 1$ (with seven channels), much lower than that of CAH (11.9%) and PH (27.9%), while at $\gamma = 2$ in the extended range (with 11 channels), TMBH still has a very low CV value of 6.5%, very close to the CV value of the pseudo incoherent case (3.2%), in striking contrast to the cases of CAH (20.7%) and PH (33.9%).

Furthermore, the example of the winter sport icon is verified in the experiment. To demonstrate the effectiveness of the TMBH method in an experiment, we developed three prototypes of OAM holographic displays (see Supplementary Note 12). First prototype implements the TMBH method, based on a DMD to time-sequentially upload OAM-multiplexed holograms. The second prototype employs the PH method as a baseline comparison, using an SLM with phase-only OAM-multiplexed holograms. The third prototype employs the CAH method as a baseline comparison, using two SLMs (an amplitude-only one and a phase-only one) for complex-amplitude modulation. In the experiment, eleven icons are encoded separately into eleven OAM

channels with the helical mode index ranging from −5 to 5 at an interval of $\Delta l = 1$ that for each of OAM channels, 50 binary OAM-multiplexed holograms are generated and time-sequentially uploaded onto a DMD at 10 kHz, following the procedure of Fig. 5. To further improve the OAM selectivity, a fundamental mode filtering aperture array is added during post-processing to rule out high-order OAM mode pixels with doughnut-shaped intensity distributions, with the post-processed results shown in Supplementary Note 13 (raw data is demonstrated in Supplementary Figs. 24 and 25).

In the experimentally reconstructed results of TMBH in Supplementary Note 13, the structure of the OAM mode pixel at each pixel is distinctive and stable, making the post-processing results clear and almost crosstalk-free. In contrast, the reconstructed results of CAH and PH contain a huge number of speckles due to inevitable interference, leading to strong crosstalk in the post-processed results. Furthermore, we verify the operating condition beyond the sampling limit ($\gamma = 2.5$) in the experiment by scaling the multiplexed channels to 17 while maintaining the image resolution, as shown in Supplementary Fig. 26. We also experimentally verify that TMBH can significantly reduce the multiplexing crosstalk at the sampling limit ($\gamma = 1$), and thus significantly improve the image quality with respect to PH method (seven channels in Supplementary Fig. 27). Besides, Supplementary Fig. 28

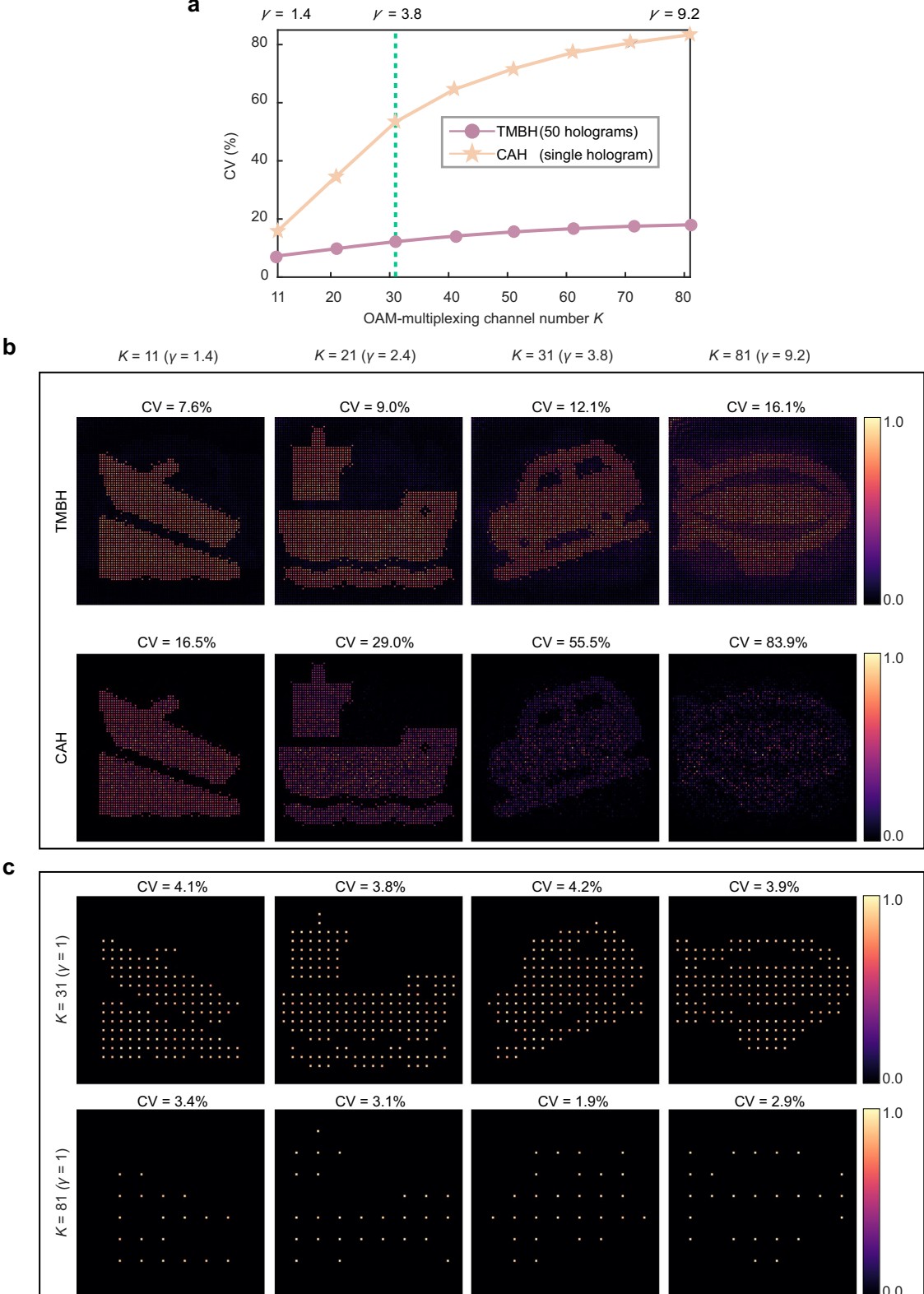

**Fig. 7 | Super-resolution reconstruction of binary images by TMBH method.**
**a** Intensity fluctuations of reconstructed images versus increasing OAM-multiplexing channel number. **b** Comparison between TMBH and CAH methods in terms of reconstruction quality at the same resolution, operating far beyond the sampling criterion limit. **c** Reconstruction results of the CAH method at the resolution limit of high capacity ($\gamma = 1$, $K = 81$).

shows the reconstructed images of sparsely sampled letters and numbers using 33 channels ($\gamma = 2$).

For the second example, the sampling condition is significantly pushed to $\gamma = 9.2$, while a series of different $800 \times 800$ binary images are separately encoded by up to 81 OAM channels with $\Delta l = 1$. Figure 7a, b indicate the potentials of capacity scaling with fixed resolution ($L = 10 \lambda/\text{NA}$) that as the channel number $K$ increases from 11 to 81, the TMBH method always exhibits high-quality images with CV value

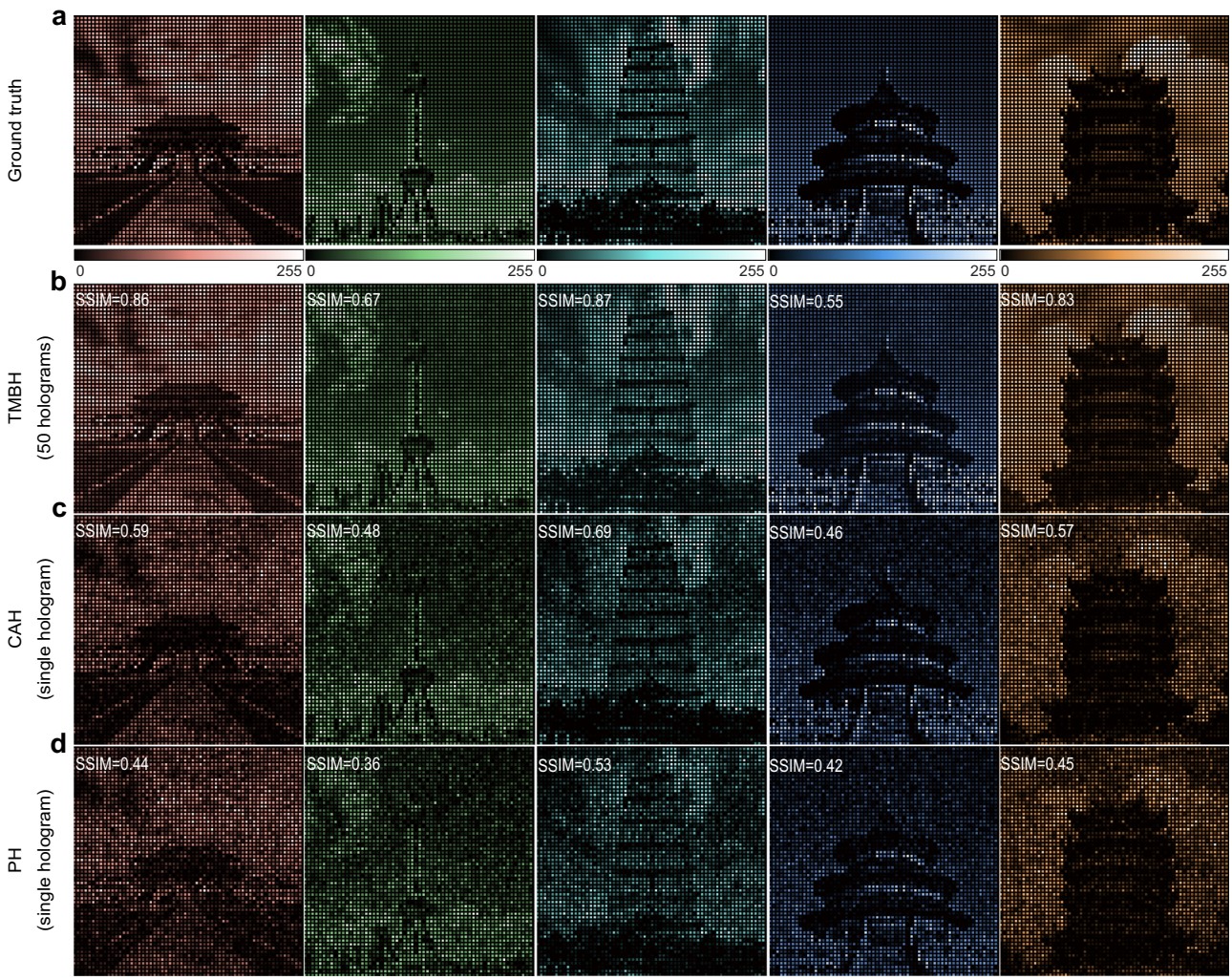

**Fig. 8 | Numerical reconstruction results of densely sampled grayscale images by three classes of OAM holography. a** Ground truth for densely sampled grayscale images. **b** Reconstructed results of the TMBH method. **c** Reconstructed results of the CAH method. **d** Reconstructed results of the PH method.

slightly rising from 7.6 to 17.6%, while CAH method fails to recover the object, having CV dramatically going up to 55.5% at $\gamma = 3.8$ and 80.0% at $\gamma = 9.2$. Notably, the super-resolution capability of the presented method over the conventional resolution limit of OAM holography is clearly shown by comparing the reconstructed results of the TMBH method ($\gamma = 9.2$, $K = 81$) in Fig. 7b (top row) and CAH method ($\gamma = 1$, $K = 81$) in Fig. 7c (bottom row), demonstrating a significant resolution enhancement by 9.2 times (from $L = 92\,\lambda$/NA to $L = 10\,\lambda$/NA) at 81 OAM channels. The experimental reconstruction results are presented in Supplementary Note 14.

The third example targets the gray images with large grayscale, which are inherently far more sensitive to intensity fluctuation than binary images. The images chosen are five gray figures of famous landmarks in China ("Forbidden City", "Oriental Pearl Tower", "Giant Wild Goose Pagoda", "Temple of Heaven", and "Yellow Crane Tower"), each with a grayscale of 256 and a resolution of $300 \times 300$. The decoding results of the OAM-multiplexed hologram are shown in Fig. 8, with the mode index of $l = 2, 1, 0, -1, -2$. In this instance, the target images are sampled by a denser array ($L = 6\,\lambda$/NA), leading to large intensity fluctuations and strong speckle noises in the post-processed decoded images of PH and CAH methods in Fig. 8. In contrast, the TMBH method faithfully reconstructs the ground truth, demonstrating a significant improvement in image quality of decoded results, as denoted by SSIM values. Notably, while keeping the high reconstructed resolution unchanged, the TMBH method allows further

enhancement of the multiplexing capacity by 2.6 times, expanding the numbers of OAM channels and reconstructed images from 5 ($\gamma = 2$) to 13 ($\gamma = 5$), as shown in Supplementary Fig. 23.

Furthermore, the example of densely sampled grayscale is experimentally investigated. As the incident OAM beam changes ($l = -2, -1, 0, 1, 2$), the corresponding five different images of landmarks encoded are displayed in turn and captured by the camera, while the post-processed experimental results are shown in Fig. 9. The SSIM values of experimental results by TMBH method are very consistent with the simulated results. One can see the TMBH method faithfully reconstructs each of the high-resolution images with 256 gray levels, clearly exhibiting the local structures and boundaries of buildings, while in the results of the PH method, targets are buried in heavy noise and can hardly be recognized. The experiment results demonstrate the strong capability of the TMBH method to recover dense gray images, which remarkably satisfies the urgent needs of OAM-based holography systems towards practical scenarios.

Finally, we illustrate the potential of TMBH for applications in holographic video display (Fig. 1). We extracted 201 images with 256 grayscale and $7600 \times 7600$ resolution from an original video of "aircraft landing". All images are sampled with $L = 38\,\lambda$/NA and encoded into 201 OAM channels ranging from −100 to 100 at the interval of $\Delta l = 1$, so that one can only reconstruct the real video stream when the 201 OAM modes are incident on the hologram in the correct sequence, while the playback content and playing frame rate can be accurately

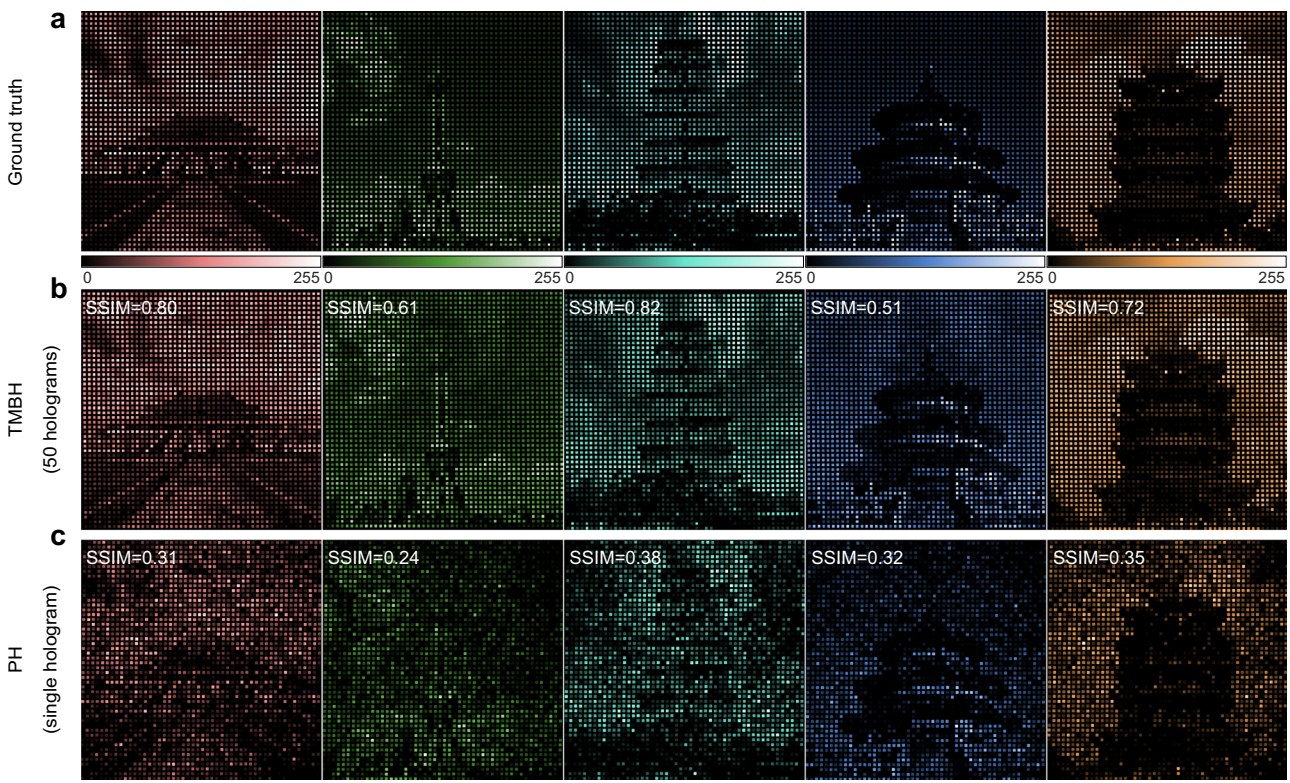

**Fig. 9 | Experimental demonstration of densely sampled grayscale reconstructed images of PH and TMBH methods. a** Ground truth for densely sampled grayscale images. **b** Reconstructed results of the TMBH method. **c** Reconstructed results of the PH method.

controlled by switching OAM modes (Fig. 1d). Figure 1e compares the SSIM values of a reconstructed frame by TMBH and CAH methods. Note that the original quality of CAH at a few channels is superior to that of TMBH, since the TMBH method that employs amplitude binary holograms has inherent binary noise. However, the SSIM of the TMBH method remains almost unchanged at 0.47–0.48 against the growing number of OAM channels within the range of $\gamma \leq 2$, beyond which it decreases very slowly, leading to an SSIM of 0.41 at 201 channels ($\gamma = 4.7$). By contrast, the SSIM value of CAH method that carries the same content decreases dramatically by 56%, from 0.54 to 0.24, as the channel number increases from 21 to 201. Figure 1e shows the fine details in two reconstructed frames of the 201-channel-multiplexing hologram of the TMBH method (see also Supplementary Movie 1), while the results of the CAH method, accompanied by strong scattering noise, are difficult to distinguish (see also Supplementary Movie 2). Comparison between Fig. 1e, f vividly indicates the superiority of TMBH in terms of high-quality, high-capacity holographic display, at a significantly enhanced resolution beyond the resolution limit of conventional OAM holography. Besides, we demonstrate that the TMBH method can be applied in three-dimensional holography, as shown in Supplementary Note 15.

## Discussion

This work has elaborated that multiplexing crosstalk in OAM holography originates from two types of interferences, and claimed that the sampling criterion limit of OAM holography can be released by effectively suppressing the coherence and thus the crosstalk in reconstructed images, thereby breaking the conventional resolution limit and capacity limit. In terms of super-resolution reconstruction, we have demonstrated, theoretically and experimentally, high-quality OAM holography towards binary images, gray images, and video display, having exceptional resolution enhancement by several times. In terms of capacity boost, the TMBH method, as well as the pseudo

incoherent concept, not only expands the largest addressable OAM index that $d_{max}$ can be larger than $L$ by an order of magnitude, but also allows OAM channels closely packed with dense index interval that all the high-quality results presented have $\Delta l = 1$, thereby leading to two-fold improvement in capacity, compared with previous demonstrations. In the experiment, we have achieved the multiplexed images with the maximum dot per inch (DPI) of 95 pixel/inch, while the number of multiplexed images is 45, reaching the super-resolution that is 5.6 times the resolution of conventional OAM holographic reconstructed images which has the DPI of only 17 pixel/inch under the same conditions. Further increasing the NA of an experimental system can improve the resolution of the reconstructed image, that can reach 1446 pixel/inch, for example, by increasing the NA from the current value of 0.02 to 0.3, which can be achieved by expanding the hologram size or reducing the focal length of Fourier lens.

Temporal multiplexing, rather than a partially coherent light source, is used to reduce the coherence of the reconstructed images, well-preventing damage in the structure of holographic images and OAM orthogonality. In addition, the computation process of binary OAM holograms is non-iterative; thus, the computation time is significantly reduced compared with the PH method (Supplementary Fig. 29). The presented approach is also compatible with other DoFs of light, achieving further enhancement in the multiplexing capacity while maintaining high reconstruction resolution.

Note that 50 pieces of binary holograms are multiplexed in time series for each object in this work; however, ten pieces is actually sufficient for suppressing the crosstalk, which is evident by the rapid decline of CV value in Supplementary Fig. 11c. Thus, TMBH method allows a holographic display rate of up to 2 kHz, using a commercial DMD operated at 20 kHz. Despite that our proof-of-principle experiments use DMD, which enjoys powerful programmability, low cost, and easy adaptiveness, the TMBH method is applicable to all types of reconfigurable holographic media. In a step towards ultrafast

switching of image frames, the presented method can be potentially implemented by fully reprogrammable and actively tunable meta-surfaces that are favored with very fast reconfiguration time (<33 ns) and compact size[39], keeping in mind that existing individually addressable metasurfaces which enable only 2-bit encoding at maximum[40–42] can operate well with binary OAM holograms. In addition, using a laser with tunable spatial coherence[43–45] as the illumination source is beneficial to eliminate the inherent binary noise of binary holograms (Supplementary Note 9) and further enhance the reconstruction quality. Besides, the pseudo incoherent concept proposed might come true, by elaborately regularizing the directions of spatial frequencies of all incoherent incident vectors into coincidence, possibly with the help of machine-learning-based methods. This work opens up new perspectives for OAM holography, and illustrates a practical and feasible solution to high-resolution, high-quality, high-volume holographic systems, paving the way for performance improvement in a wide range of applications, including smart display, holographic encryption, augmented reality, and optical storage.

## Methods

### Coefficient of variation
The coefficient of variation (CV) is defined as the ratio between the standard deviation of intensity fluctuations and the average intensity in the evaluated area:

$$CV = \frac{\sqrt{\frac{1}{PQ}\sum_{p=1}^{P}\sum_{q=1}^{Q}(I_{p,q} - \bar{I})^2}}{\bar{I}} \tag{1}$$

As an evaluation parameter that does not require a true value, CV is often used to evaluate the speckle contrast in images[45]. In this work, CV is used to evaluate the reconstructed results of binary images. A larger CV value represents a higher level of intensity fluctuation in the binary image, which suggests the image quality is getting worse.

### Evaluation of image quality
To evaluate complex images with multiple grayscales, we introduce structural similarity index measure (SSIM)[46] as an objective image quality evaluation function, which fully takes advantage of known characteristics of the human visual system (HVS). SSIM is generated by combining five functions:

$$SSIM = \frac{2(\mu_e\mu_t + c_1)(2\sigma_{e,t} + c_2)}{(\mu_e^2 + \mu_t^2 + c_1)(\sigma_e^2 + \sigma_t^2 + c_2)} \tag{2}$$

where $\mu_t$ and $\mu_e$ are the mean values of the ground truth and evaluated images, respectively. $\sigma_t$ and $\sigma_e$ are the standard deviations of the ground truth and evaluated images, respectively. $\sigma_{e,t}$ is the covariance between two images, $c_1$ and $c_2$ are two constants. The SSIM takes values between 0 and 1, while a larger SSIM means that the structure of the evaluated image has higher similarity with the ground truth.

## Data availability
The Source data are available from the corresponding author upon request. All data needed to evaluate the conclusion are present in the main text and/or the Supplementary Information.

## Code availability
Codes used for this work are available from the corresponding author upon request.

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

## Acknowledgements

X.F. acknowledges funding support from the National Natural Science Foundation of China (61975087). We thank Prof. Liangcai Cao, and Prof. Hao Zhang in Tsinghua University, and Prof. Xingpeng Yan in Army Academy of Armored Forces for their technical assistance with the experiment and useful discussions.

## Author contributions

Z.S. and X.F. proposed the idea and conceived the experiment. Z.S. performed the theoretical calculations. Z.S. constructed the experiment with the help of Z.W., Z.Z., and K.L., acquired the data, and carried out the data analysis. Z.S., X.F., and Q.L. completed the writing of the paper.

## Competing interests

The authors declare no competing interests.
