## [Peer Review File · Nature Communications]

Super-resolution orbital angular momentum holographyREVIEWER COMMENTS

Reviewer #1 (Remarks to the Author):

In this manuscript, the authors propose a comprehensive theoretical model of same pixel interference and adjacent pixel interference in orbital angular momentum (OAM) holography. Through the temporal multiplexing techniques, they have demonstrated high quality and high resolution OAM holography. The rationale and results are original. The data, rational and descriptions are clear. Before I can recommend it for publication, the authors may consider the following suggestions to improve the technical soundness of the paper:

1. In my understanding, temporal multiplexing in Fig. 2d is akin to adding a grating function on the incident k-vector in the temporal domain. Right? The author should discuss more about Fig. 2d.
2. As shown in Fig 3a, the OAM beams with $l=1$ is illuminated on the hologram. What is amplitude distributions of the incident OAM beam? Would it affect the intensity fluctuations of the reconstructed images for both the coherent and pseudo inherent cases?
3. In Fig. 4, the author only considers the 3×3 spot array with the same intensity. What will happen for the spot array with different intensities?
4. In the second row of Fig. S5(b), we can see the pattern "B" dominated by SPI and API has a very strong background noise. I understand the low intensity fluctuations due to pseudo incoherent interference. But I suggest the author to explain how to achieve the ultrahigh resolution OAM resolution with a strong background noise, especially for multiplexing the grey-level images.
5. In Fig. 7c and Fig. 10, the authors should not compare the images reconstructed by a single hologram in CAH or PH with 50 holograms in TMBHs. The holograms have different spatial bandwidth products.
6. In Fig. 10, I cannot see the SSIM as shown in Fig. 8. And the author should discuss the relatively poor experimental results in TMBH.
7. I suggest the author the discuss the maximum dot per inch (DPI) in the discussion part.
8. Could the TMBH be applied in three dimensional (3D) holography? I suggest the author to give a discussion.

Minor:

1. In the manuscript. "The temporal multiplexing OAM holography (TMBH) ", the word "Binary" is missing.
2. The author should consider put some main figures in the supplementary material.

Reviewer #2 (Remarks to the Author):

The authors have proposed to use temporal multiplexing technique to improve the conventional resolution limit of OAM holography. They claimed that the technique has eliminated the existing multiplexing cross-talks and decreased the sampling constraints. However, the experimental demonstration was realized based on conventional optical elements (i.e., SLM and DMD), not on high resolution optical metasurfaces. I cannot make a decision about this manuscript since the novelty and uniqueness need to be further clarified. In addition, several suggestions are given below to improve the manuscript.

- 1) The term "super-resolution" is used several times in the manuscript and should it be defined for better understanding? What is the benchmark for super-resolution in the case of holography? If we compare the number of pixels per inch (ppi) of the reconstructed image (presented in the manuscript) with the conventional super-resolution images, the difference is significantly large. Therefore, authors should address this point to avoid any confusion.

Furthermore, on page 2, line 76, the authors state that "...the OAM-addressed 201 frame video display (with the grayscale of 256) at a super-resolution of 200×200 pixels." Are these " 200×200 pixels" for one frame or ppi. Authors should also explain this in the manuscript.

2) With reference to lines 114-116 on page 4, the authors have mentioned that "... However, this case is practically challenging to realize due to the violation of the Bragg diffraction formula, thus a coherent analogy to that is alternatively carried out in this work, by temporal multiplexing of coherent beam thereby suppressing the coherence...".

What is the reason for carrying out a comparison between coherent and pseudo incoherent case for same pixel interference (SPI) and adjacent pixel interference (API), if the pseudo incoherent case is a violation of Bragg diffraction formulae? The more practical scenario, i.e., temporal multiplexing should have been considered.

3) What is the super-resolution condition mentioned on Page 4, line 120?

4) The definition of API interference should be rephrased for better readability.

5) The intensity fluctuations of the reconstructed images for numerical results are provided in Figure 7a. It is recommended to incorporate experimental results as well. Moreover, authors should also include experimental results in Figure 6 for the sake of comparison.

6) What is the OAM purity in each channel/frame? For example, in Fig. S7, while addressing the reconstructed results of temporal multiplexing binary OAM holography, the intensity profile of OAM at different moments (T1-T50) cannot be recognized clearly.

Authors should include the discussion on the amount of OAM content at the each moment. See the following reference for more details:

a) Sroor, Hend, et al. "High-purity orbital angular momentum states from a visible metasurface laser." *Nature Photonics* 14.8 (2020): 498-503.

7) Overall writing and organization of the manuscript should be improved.

Response to the reviewers and editor

We would like to thank all of the reviewers their careful reading of our work and useful comments.

Reviewer #1

Comment 1.1:

“In this manuscript, the authors propose a comprehensive theoretical model of same pixel interference and adjacent pixel interference in orbital angular momentum (OAM) holography. Through the temporal multiplexing techniques, they have demonstrated high quality and high resolution OAM holography. The rationale and results are original. The data, rational and descriptions are clear. Before I can recommend it for publication, the authors may consider the following suggestions to improve the technical soundness of the paper”

Response: Many thanks for the positive, warm comments of referee on our work. We address all the insightful technique queries in the following that greatly help us to improve the manuscript and clarify some important points.

Comment 1.2:

“1. In my understanding, temporal multiplexing in Fig. 2d is akin to adding a grating function on the incident k -vector in the temporal domain. Right? The author should discuss more about Fig. 2d.”

Response: Thanks for this insightful comment. We fully agree with the referee that “*temporal multiplexing in Fig. 2d is akin to adding a grating function on the incident k -vector in the temporal domain*”.

Temporal multiplexing is indeed the addition of a phase grating in the temporal domain, and can be interpreted as a superposition of two sub-processes (see Fig. R1). The phase grating generates n incoherent wave vectors in the temporal domain, the number of which is equal to the number of multiplexed holograms. Each wave vector incident on the hologram produces an independent diffraction pattern. The observed reconstructed image consists of m diffraction patterns superimposed in an intensity manner, where m is determined by the duration Δ of each diffraction pattern and the observation time duration T .

Fig. R1 Detailed illustration of the temporal multiplexing case. a Coherence suppression with temporal multiplexing. **b** A wave vector is incident on a phase grating in the temporal domain and generates n incoherent wave vectors. **c** The n incoherent wave vectors produce n independent diffraction patterns. It is worth noting that the n incoherent wave vectors of Fig. R1c have the same direction with respect to Fig. 2b, because the n incoherent wave vectors originate from the temporal domain expansion of the same wave vector.

Action Taken:

In the revised version, we have discussed Fig. 2d in more details in Supplementary Note 2.

Comment 1.3:

“2. As shown in Fig 3a, the OAM beams with $l=1$ is illuminated on the hologram. What is amplitude distributions of the incident OAM beam? Would it affect the intensity fluctuations of the reconstructed images for both the coherent and pseudo inherent cases?”

Response: Thanks for asking. In Fig. 3a, the incident OAM beam is actually a theoretical spiral phase with the amplitude distribution as a simple circular aperture function (see Fig. R2).

Fig. R2 The amplitude and phase of the incident OAM beam in Fig. 3a.

As shown in Fig. R2, the square discrete area size is L and the diameter of the circular aperture function is D . In Fig. 3a, we set $D = 0.5 L$. According to the referee’s suggestion, we investigate how the value of D affects the intensity fluctuations of the reconstructed image for both the coherent

and pseudo incoherent cases, as shown in Fig. R3.

Fig. R3 Intensity fluctuations of the reconstructed image versus the circular aperture size D .

For the coherent case, the intensity fluctuations of reconstructed images increase as the aperture diameter D grows for two reasons. First, the increased D and the corresponding reduced spot size in the Fourier plane enlarge the light intensity at the spot center, thereby increasing the intensity fluctuations of each pixel location (CV goes up from 8.0% to 12.3%). Second, the diameter of filtering aperture array does not precisely match the circular aperture, due to discrete errors that the filter aperture is discretized into a two-dimensional matrix and cannot be continuously adjusted in size. In the pseudo-coherent case, the intensity fluctuations of the reconstructed images hardly change (CV from 2.7% to 3.9%) with the diameter of the aperture function.

Comment 1.4:

“3. In Fig. 4, the author only considers the 3*3 spot array with the same intensity. What will happen for the spot array with different intensities?”

Response:

Thanks for the suggestion. To investigate the case for the spot array with different intensities, we set random intensity values (uniformly distributed in the range of 0 to 1) for the 8 pixels around the center pixel and run 100 trials to obtain the results as shown in Fig. R4.

The case depicted in Fig. R4a is actually the API model when grayscale images are multiplexed. For the coherent case, it shows a slightly smaller level of intensity fluctuation at $\gamma = 2.9$ (CV=30%), compared with Fig. 4c with spot array having the same intensity (CV=36%). For the pseudo-incoherent case, CV becomes nonzero, reaching 1% at $\gamma = 2.9$, due to the fact that intensity of the pixels around the center one vary at each trial, thereby introducing a little intensity fluctuation. For non-signal location, Fig. R4b shows a decrease in the average intensity of noise at the center pixel (2.5% of signal intensity at $\gamma = 2.9$) compared with that in Fig. 3e (6.0% of the signal intensity).

Fig. R4 API model for the spot array with random intensities. **a** Fluctuations of intensity integral versus γ , for the API effect on a signal location, with densely packed OAM modes ($l=0, 5, 10, 15$) at nine adjacent pixels. The CV value indicating the intensity fluctuations is calculated from 100 trials. Specific intensity results of each trial are shown for the cases of $\gamma=0.96$ and $\gamma=2.90$ (right

column). **b** Fluctuations of intensity integral in d versus γ , for the API effect on a non-signal location, with densely packed OAM modes ($l=5, 10, 15$) at nine adjacent pixels.

Action Taken:

In the revised version, we have studied the API effect of spot array with different intensities in Supplementary Note 6.

Comment 1.5:

“4. In the second row of Fig. S5(b), we can see the pattern “B” dominated by SPI and API has a very strong background noise. I understand the low intensity fluctuations due to pseudo incoherent interference. But I suggest the author to explain how to achieve the ultrahigh resolution OAM resolution with a strong background noise, especially for multiplexing the grey-level images.”

Response: Thanks for the comment and suggestion.

First of all, it should be emphasized that the final extracted image is the one after the filtering aperture array, so we only need to pay attention to the intensity of the background noise inside the filtering aperture, while the noise distributed outside the aperture array does not have any effect on the extracted image. The background noise in the pseudo incoherent case originates from the intensity superposition of adjacent OAM pixels (API).

For the pseudo incoherent case, the background noise intensity within the filtering aperture is directly added on the signal, causing an offset in the signal intensity with respect to the ground truth. Taking the rightmost case ($\gamma=2.5$) in Fig. S5b as an example, the intensity of the adjacent OAM pixels within the filtering aperture has been analyzed in detail in Fig. S4b, and the noise within the filtering aperture is approximately equal to 30% of the signal intensity when $\gamma=2.5$.

Figure R5 compares the binary image and grayscale image after adding the background noise. It can be seen that the effect of background noise is not significant for binary images, while the contrast of gray images decreases with the addition of background noise but the images are still distinguishable. It is necessary to emphasize that Fig. R5 shows an extreme case, because the larger the $|l_{\max}|$, the weaker the noise in the filtering aperture, as the relative intensity of the adjacent OAM pixels is weaker, for the same γ condition. Moreover, in practice, not all adjacent pixels contribute to the background noise, thus the quality of the reconstructed images is often higher than the case illustrated in Fig. R5 (see also Fig. 10 and Fig. S5b). As a comparison, the reconstructed images in the coherent case under the same condition (Fig. R5) has strong intensity fluctuations, which completely destroys the original grayscale information of the image.

In conclusion, the key of achieving ultra-high OAM resolution is to attenuate the intensity fluctuations of the reconstructed image, while the background noise, which represents a constant image offset, has little effect on the quality of the reconstructed image when the multiplexed number is relatively small.

Fig. R5 Example of adding background noise to binary and gray images.

Comment 1.6:

“5. In Fig. 7c and Fig. 10, the authors should not compare the images reconstructed by a single hologram in CAH or PH with 50 holograms in TMBHs. The holograms have different spatial bandwidth products.”

Response: Thanks for the comment and suggestion. As mentioned by the referee, the numerical and experimental results presented in this paper include reconstructions of single hologram of CAH and PH, as well as 50 holograms of TMBH. According to the definition of spatial bandwidth product (SBP), the SBP of CAH results is twice as large as that of PH, and the SBP of TMBH is 50 times as large as that of PH. However, the temporal multiplexing method in this paper achieves low crosstalk images, exactly by extending the SBP, thus the higher SBP is an advantage of our method. In our experiments we use a high refresh rate modulation device (DMD), which allows us to accumulate 50 times the SBP of the phase hologram in a very short time ($t = 1/200$ s).

Finally, we would like to emphasize that single hologram approach has the advantage of obtaining reconstruction at high speed, while the temporal multiplexing result of 50 holograms can only be accurately obtained at exposure times greater than $1/200$ s, which is the disadvantage of TMBH method. Moreover, in the revised version, we have labeled “single hologram” for each result of CAH and PH method, and labeled “50 holograms” for TMBH results to avoid confusion.

Comment 1.7:

“6. In Fig. 10, I cannot see the SSIM as shown in Fig. 8. And the author should discuss the relatively poor experimental results in TMBH.”

Response: Thanks for the suggestion. As the referee points out, the experimental results shown in Fig. 10 have relatively poor image quality compared to the simulation results shown in Fig. 8, to which we have added the corresponding discussion.

Compared with the simulation results, the reconstructed quality in the experiment is degraded mainly in two aspects, as shown in Fig. R6. Firstly, the gray values of the upper part of the images are significantly higher than those of the lower part, due to the fact that the upper part, which is closer to the zero-order diffraction level, has stronger diffracted intensity. Secondly and more crucially, there exists intensity crosstalk among the five multiplexed images, as manifested in Fig. R6 by the undesirable introduction of image content from other OAM channels. The main reason for this phenomenon is that the aberrations in the optical system in the experiment (mainly in the form of astigmatism) cause a slight off-centering of OAM mode in the Fourier plane (the plane where the reconstructed image is located), thus the filtering aperture array cannot completely filter out the image information from other channels. It is easily understood that the intensity crosstalk is more obvious in the upper half of the image, due to the higher light intensity as stated above.

Fig. R6 Schematic diagram of the intensity crosstalk of the reconstructed image obtained in the experimental results of TMBH method.

Notably, during the revision stage, we have well corrected the optical distortions in the experimental setup and pre-compensated the diffraction efficiency at different locations of the images. Compared with the results of submitted original version, the intensity crosstalk and intensity inhomogeneity problems described above are efficiently solved, therefore the updated experimental results of TMBH as presented in Fig. R7 have the SSIM values very close to those in the simulation.

Fig. R7 Experimental demonstration of densely sampled grayscale reconstructed images of PH and TMBH methods.

Action Taken:

In the revised version, we have updated the experimental results of TMBH with higher quality, and have added the metric of SSIM.

Comment 1.8:

“7.1 suggest the author the discuss the maximum dot per inch (DPI) in the discussion part.”

Response: Thanks for the suggestion. We have achieved the multiplexed images with the maximum DPI of 95 pixel/inch in the experiment, while the number of multiplexed images is 45, reaching the super-resolution 5.5 times the resolution of conventional OAM holographic reconstructed images which has the DPI of only 17 pixel/inch under the same conditions.

Further increasing the NA of experimental system can improve the resolution of reconstructed image. For example, the resolution of reconstructed image can reach 1446 pixel/inch, by increasing the NA from current value of 0.02 to 0.3, which can be achieved by expanding the hologram size or reducing the focal length of Fourier lens.

Action Taken:

In the revised version, we have added the discussion of DPI in the discussion part.

Comment 1.9:

“8. Could the TMBH be applied in three dimensional (3D) holography? I suggest the author to give a discussion.”

Response: Thanks for the suggestion. In the revised version, we illustrate by two examples that TMBH can be applied in 3D holography.

The first example is a 3D OAM-selective hologram, the idea of which was previously illustrated in the paper [Nat. Photonics 14, 102-108 (2020)]. Here the reconstructed results of 3D OAM-selective hologram by our work are shown in Fig. R8. In the plane of $z=400$ mm, an incident OAM beam with helical mode index $l=1$ can reconstruct the image of the Big Ben, while the other image (the Eiffel Tower) is out of focus in this plane. In the plane of $z=450$ mm, an incident OAM beam with helical mode index $l=2$ reconstructs the image of the Eiffel Tower, while the image of the Big Ben is out of focus.

The second example is a 3D OAM-multiplexing hologram, as previously presented in the paper [Nat. Nanotechnol. 15, 948-955 (2020)]. Here we select OAM beams with helical mode indices ranging from -4 to 4 to sequentially illuminate a 3D OAM-multiplexing hologram, resulting in a total of 10 images reconstructed in two different planes, as shown in Fig. R9.

Fig. R8 Experimental demonstration of using an OAM-selective hologram for the reconstruction of three-dimensional OAM-dependent holographic images. **a** The experimentally reconstructed holographic images using an incident OAM beam with a helical mode index of $l = 1$. **b** The experimentally reconstructed holographic images using an incident OAM beam with a helical mode index of $l = 2$.

Fig. R9 Experimental demonstration of using an OAM-multiplexing hologram for the reconstruction of three-dimensional OAM-dependent holographic images. **a** Reconstructed images from an OAM-multiplexing hologram at a reconstruction distance of $z=400$ mm. **b** Reconstructed images from an OAM-multiplexing hologram at a reconstruction distance of $z=450$ mm.

Action Taken:

In the revised version, we have added the discussion of TMBH applied in 3D holography in Supplementary Note 15.

Comment 1.10:

“In the manuscript. “The temporal multiplexing OAM holography (TMBH) “, the word “Binary” is missing.”

Action Taken: Thanks for the reminder. We have corrected the typo in the revised version.

Comment 1.11:

“The author should consider put some main figures in the supplementary material.”

Action Taken: Thanks for the suggestion. We have put original Fig. 9 in the supplementary material.

Reviewer #2

Comment 2.1:

The authors have proposed to use temporal multiplexing technique to improve the conventional resolution limit of OAM holography. They claimed that the technique has eliminated the existing multiplexing cross-talks and decreased the sampling constraints. However, the experimental demonstration was realized based on conventional optical elements (i.e., SLM and DMD), not on high resolution optical metasurfaces. I cannot make a decision about this manuscript since the novelty and uniqueness need to be further clarified. In addition, several suggestions are given below to improve the manuscript.

Response: Thank you very much for taking the time to offer an enthusiastic review of our work. We have answered all your technical questions below, which greatly help us to enhance the impact of this work.

First, and very importantly, the innovative points of our paper are stated as follows.

- (1) We systematically analyze, for the first time, the multiplexing crosstalk in OAM multiplexing holography, i.e., we propose two interference models (SPI and API), and reveal the origin of noise in OAM multiplexing holography.
- (2) We break the resolution limit in OAM multiplexing holography for the first time, by using temporal multiplexing binary OAM holograms (TMBH), and achieve a reconstructed image with a resolution 9.2 times that of conventional OAM holography in simulation, and a resolution 5.6 times that of conventional OAM holography in experiment.
- (3) We achieved OAM multiplexing of grayscale images for the first time, in both simulation and experiment, and the reconstruction results are quite good.

Regarding the use of conventional optical elements (i.e., SLM and DMD), instead of high resolution optical metasurfaces.

- (1) The main contribution in this paper is the analysis and substantial reduction of crosstalk in OAM multiplexed holography. In this paper we have shown that the key of reducing the crosstalk noise is to reduce the coherence in the reconstructed image, and the purpose of this paper is to use a method

(temporal multiplexing) that can break the resolution limit in OAM multiplexing holography and obtain higher quality reconstructed images. Therefore, this paper focuses on the method breakthrough rather than the device.

(2) Due to current experimental conditions, we do not have the ability to design and process high-resolution metasurfaces, so we use conventional optical devices (SLM and DMD) in our experiments. And the temporal multiplexing method requires us to use programmable optics, while the resolution of programmable metasurfaces are currently relatively low (for details, see Refs. [Badloe, Tunable Metasurfaces: The Path to Fully Active Nanophotonics], [He, Q., Sun, S., & Zhou, L. Tunable/Reconfigurable Metasurfaces: Physics and Applications. Research 2019, 1849272 (2019).]).

(3) We have discussed in Discussion that in the future, as the technology of programmable metasurfaces becomes sophisticated, the implementation of TMBH using programmable metasurfaces will further improve the resolution and image quality of the reconstructed images.

Comment 2.2:

“1. The term “super-resolution” is used several times in the manuscript and should it be defined for better understanding? What is the benchmark for super-resolution in the case of holography? If we compare the number of pixels per inch (ppi) of the reconstructed image (presented in the manuscript) with the conventional super-resolution images, the difference is significantly large. Therefore, authors should address this point to avoid any confusion.”

Response: Thanks for the comment and suggestion.

Firstly, it is important to state that the concept of super-resolution in this paper applies only to OAM holography. The resolution limit of OAM holography is defined in this work, based on three seminal articles on OAM holography [Nat. Commun. 10, 2986 (2019); Nat. Photonics 14, 102-108 (2020); Nat. Nanotechnol. 15, 948-955 (2020)], i.e., the distance L between two pixels needs to be always larger than the diameter of the maximum addressable OAM mode (d_{\max}), which is similar to the concept of the conventional resolution limit (the distance between two pixels needs to be always larger than the Rayleigh criterion). For presentation convenience, we define a factor ($\gamma=d_{\max}/L$). According to the definition, the case of $\gamma=1$ represents the OAM holography at resolution limit, and the cases of $\gamma>1$ represent the super-resolution OAM holography as studied in this paper.

To avoid confusion, in the revised version we have made a comparison between the resolution limit in a conventional optical system and the resolution limit in OAM holography. The main interest of the conventional resolution limit lies at the image plane of the optical system, where the distribution of the point spread function is usually the diffraction pattern of the circular aperture (Bessel function), and the factors affecting the resolution limit are the light wavelength and the numerical aperture (NA) of the optical system. In contrast, the resolution limit in OAM holography is examined at the back focus plane (Fourier plane) of the lens, where the point spread function is the distribution of the maximum addressable OAM mode in the reconstructed image. The resolution limit of OAM holography depends on the helical index of the maximum addressable OAM mode, in addition to the optical wavelength and the NA of the Fourier holographic system (see Fig. S1). Furthermore, we have added Table R1 to illustrate more clearly the connection and difference between the two resolution limits.

Table R1 Comparison between the resolution limit in a conventional optical system and the resolution limit in OAM holography.

	Main location	Point spread function	Affecting factors
Conventional resolution limit	Image plane of the optical system	The diffraction pattern of the circular aperture	Wavelength and NA
Resolution limit in OAM holography	Back focus plane (Fourier plane) of the lens	The distribution of the maximum addressable OAM mode	Wavelength, NA and l_{\max}

Action Taken: In the Supplementary Note 1 of revised version, we have explained in detail the concept of super-resolution in OAM holography, compared with the concept of the conventional resolution limit.

“Furthermore, on page 2, line 76, the authors state that “...the OAM-addressed 201 frame video display (with the grayscale of 256) at a super-resolution of 200×200 pixels.” Are these “200×200 pixels” for one frame or ppi. Authors should also explain this in the manuscript.”

Action Taken: Thanks for asking. Here “200×200 pixels” refers to the number of pixels in one frame, which has been explained in the revised version.

Comment 2.3:

“2. With reference to lines 114-116 on page 4, the authors have mentioned that “... However, this case is practically challenging to realize due to the violation of the Bragg diffraction formula, thus a coherent analogy to that is alternatively carried out in this work, by temporal multiplexing of coherent beam thereby suppressing the coherence...”

What is the reason for carrying out a comparison between coherent and pseudo incoherent case for same pixel interference (SPI) and adjacent pixel interference (API), if the pseudo incoherent case is a violation of Bragg diffraction formulae? The more practical scenario, i.e., temporal multiplexing should have been considered.”

Response: Thanks for pointing it out. We’d like to clarify the reason of introducing the pseudo-coherent case. Since the reconstructed image in the pseudo-coherent case is completely free of any interference, all coherent superpositions are replaced by intensity superpositions, and it is convenient to study the effect of coherence on the distribution and quality of the reconstructed images by comparing the reconstructed images in the pseudo-coherent case with the reconstructed images in the coherent case. In addition, the pseudo-coherent state as an ideal model gives a theoretical upper limit on the quality of the reconstructed images in OAM multiplexed holography (the theoretical minimum value of multiplexed crosstalk).

Temporal multiplexing is a coherent means of approximating the pseudo-coherent case, and its degree of approximation is related to the number of temporal multiplexing holograms, thus a direct

comparison of the reconstructed images in the temporal multiplexing case and the coherent case needs to consider the number of temporal multiplexing, which increases the complexity of the multiplexing crosstalk model.

We agree with the referee, and have compared the coherent case with the case of temporal multiplexing of 50 holograms in the Supplementary Note 7, where we have added the temporal multiplexing results for both models of same pixel interference (SPI) and adjacent pixel interference (API). In Figs. R10 and R11, we compare the temporal multiplexing case with the coherent case in terms of both SPI and API effects.

Comparing the results of Fig. R10 with those in Fig. 3, it can be concluded that the temporal multiplexing case and the pseudo-incoherent case behave consistently in the model of SPI. The intensity fluctuations of the reconstructed images for the temporal multiplexing case shown in Fig. R10c are almost the same as that for the pseudo-incoherent case. In addition, the intensity distributions of three representative pixels in the reconstructed image for the temporal multiplexing case (Fig. R10b, R10c) are extremely close to the pseudo-incoherent case (Fig. 3b, 3c).

The capability of temporal multiplexing of suppressing intensity fluctuations is illustrated in Fig. R11a, where the CV of temporal multiplexing rises to 5% under super-resolution conditions of $\gamma = 2.9$, while the CV of pseudo-incoherent case is always close to zero (Fig. 4c). The simulation and experimental results presented in this work verify that the CV of the reconstructed image of TMBH rises slowly with increasing γ .

Fig. R10 SPI model of the case of temporal multiplexing of 50 holograms. **a** OAM beams with the helical mode index $l=1$ is incident on the OAM-multiplexed hologram, and images carried by different OAM channels are displayed simultaneously (left). Performances of coherent case (top right), and temporal multiplexing case (bottom right) are compared. **b** Three representative pixels from the reconstructed image in **a**, for the coherent (left) and temporal multiplexing (right) cases. Intensity distributions are presented in both 2D profiles and 1D cross-sectional views where the values indicate the normalized intensity integral within the region as denoted by vertical dotted lines. **c** Intensities of all pixels in the post-processed results of the reconstructed image in **a**.

Fig. R11 API model of the case of temporal multiplexing of 50 holograms. a Fluctuations of intensity integral versus γ , for the API effect on a signal location, with densely packed OAM modes ($l=0, 5, 10, 15$) at nine adjacent pixels. The CV value indicating the intensity fluctuations is calculated from 100 trials. Specific intensity results of each trial are shown for the cases of $\gamma=0.96$ and $\gamma=2.90$ (right column). **b** Fluctuations of intensity integral in d versus γ , for the API effect on a non-signal location, with densely packed OAM modes ($l=5, 10, 15$) at nine adjacent pixels.

Action Taken: We have compared the coherent case with the case of temporal multiplexing of 50 holograms in the Supplementary Note 7, where we have added the temporal multiplexing results for both models of SPI and API.

Comment 2.4:

“3. What is the super-resolution condition mentioned on Page 4, line 120?”

Response: Thanks for the comment and suggestion. According to the definition of super-resolution described in response 2.2, line 120 of the paper describes the case of $\gamma=1.75$ (the rightmost case in Fig. 2d), which reaches super-resolution in OAM holography. Specifically, the multiplexed OAM modes in Fig. 2 are $l=0, 3$, and the sampling distance of multiplexed images of conventional OAM holography should be no less than $14 \lambda/\text{NA}$. But the sampling distance of the images in Fig. 2d is $8\lambda/\text{NA}$, breaking the diffraction limit of 1.75 times in OAM holography.

Comment 2.5:

“4. The definition of API interference should be rephrased for better readability.”

Response: Thanks for the suggestion. In original manuscript, we define the API as "interference between OAM channels at the pixel and those from adjacent pixels". We have rephrased the definition of API as "Interference between OAM modes in adjacent pixels" in the revised version, and have explained the API in detail as follows.

API: The reconstructed image of OAM holography is composed of a series of discrete pixels, and each pixel location contains one or more OAM modes, thus it can be considered as a kind of OAM array. When the distance between the discrete pixels is smaller than the diameter of OAM modes, the OAM modes at each pixel location will have interference and destroy the structure of the reconstructed image. We define this effect as "Interference between OAM modes in adjacent pixels", i.e., API.

Action Taken: We have rephrased the definition of API, and explained it in detail.

Comment 2.6:

“5. The intensity fluctuations of the reconstructed images for numerical results are provided in Figure 7a. It is recommended to incorporate experimental results as well. Moreover, authors should also include experimental results in Figure 6 for the sake of comparison.”

Response: Thanks for the suggestion. We have added experimental results (Fig. R12 and R13) in the Supplementary Note 13 and 14, corresponding to the numerical results in Fig. 6 and Fig. 7 respectively.

In Fig. R12 we compare in detail the performance of three types of OAM holograms: PH, CAH, and TMBH proposed in this work. The target images in Fig. R12 are identical to that of the simulation in Fig. 6 in the main text, i.e., a set of 300×300 binary images of winter sport icons are separately encoded into multiple OAM channels with a sampling distance of $L = 8\lambda / NA$, and the index interval is $\Delta l = 1$. Figures R12a and R12b show the reconstruction results for 7 and 11 OAM channels, respectively, indicating that the TMBH method obviously has the best image quality and the lowest CV of all OAM channels (see also Fig. R12c). Figure 12d shows that TMBH has a CV value of 8.1% at the conventional boundary $\gamma = 1$ (with 7 channels), which is close to the result in the simulation (5.5%), while TMBH has a CV value of 10.1% at the extended range of $\gamma = 2$ (with 11 channels), which is close to the simulated value (6.5%).

It is worth noting that the reconstruction quality of CAH in the experiment is close to PH, significantly lower than the simulated results, due to two reasons: (1) We use the theoretical complex amplitude value for CAH in the simulation, which does not take into account the quantization error, thus the reconstructed image of CAH in the simulation is an ideal value, while it is impossible to achieve such reconstruction quality in the experiment. The reason for using the theoretical complex amplitude values in the simulation is to highlight the superiority of TMBH, showing that the reconstruction quality of TMBH is even better than that of the theoretical complex amplitude holography. (2) We don't have the experimental conditions of complex-amplitude metasurfaces. In order to incorporate the experiment results of CAH as the referee requires, we carry out the CAH experiment (Fig. R12 and R13), by using two spatial light modulators (an amplitude-type one and

a phase-type one, with the experimental setup shown in Fig. R14), which however is not as effective as using complex amplitude metasurfaces for complex amplitude modulation. Complex amplitude metasurfaces can achieve both amplitude and phase modulation on a single cell, and the modulation accuracy is better than that of conventional spatial light modulators. For more details, please refer to Ref. “Complex-amplitude metasurface-based orbital angular momentum holography in momentum space, Nat. Nanotechnol. 15, 948 (2020).”

In addition, the experimental results in Fig. R13 corresponds to the numerical results in original Fig. 7. Due to the resolution limitation of the modulation device used in the experiment (device resolution is 1080×1080 , lower than the simulation resolution of 2160×2160), the resolution of the binary image used in the experiment is 400×400 . Figure R13a demonstrates the capacity scaling capability at a fixed resolution ($L = 10 \lambda/NA$) that as the number of channels K increases from 5 to 45, the TMBH method can always reconstruct the multiplexed images, having the CV value increased from 9.5% to 29.0%. In contrast, CAH method fails to recover the object at large value of K , with the CV dramatically increased from 35.8% to 83.3%. The experimental reconstruction of TMBH method exceeds the conventional resolution limit of OAM holography, showing a 5.6-fold resolution improvement (from $L=56 \lambda/NA$ to $L=10 \lambda/NA$).

Note that the experimental results of TMBH presented in Fig. R13 are inferior to the simulation results presented in Fig. 7 in two aspects: (1) The fluctuation of the reconstructed images in Fig. R13a (CV increases from 9.5% to 29.0% as K increases from 5 to 45) is higher than that in Fig. 7a (CV increases from 7.6% to 17.6% as K increases from 11 to 81). (2) The background noise of the reconstructed images in Fig. R13b (top row) is stronger than that in Fig. 7b (top row).

There are three main reasons why the experimental results have lower quality than the simulation results: (1) Each of multiplexed channels in the experiment of OAM holography takes a portion of incident intensity, thus multiplexing up to dozens of images demands a rather strong incident beam, otherwise the signal from single channel is too weak for camera acquisition. However, the increasing level of incident intensity inevitably aggravates SPI and API effects in the reconstructed images. (2) The final reconstructed images shown are obtained after the filtering aperture array in post-processing, while the filtering aperture array is calculated from the captured spot array (for more details see references: [Nat. Photonics 14, 102-108 (2020)] and [Nat. Nanotechnol. 15, 948-955 (2020)]). However, due to calculation error, the spatial location of the calculated filter aperture array may be deviated from the actual location by a few pixels, allowing light from undesirable OAM channels enter the filter aperture. This effect increases the intensity fluctuation in the experimentally reconstructed image compared with simulation, especially when multiplexing a lot of images since the light from the other OAM channels is much stronger. (3) The experimental results are inevitably affected by the modulation error of the modulation device.

Fig. R12 Experimental reconstruction results of binary images by three classes of OAM holography. **a** Reconstructed results (partially shown) with 7 OAM channels. **b** Reconstructed results (partially shown) with 11 OAM channels. **c** Quantitative comparison of reconstruction quality. **d** Evolution of intensity fluctuation of reconstructed images versus the number of OAM multiplexing channels, where the multiplexing channel numbers for $\gamma=1$ and $\gamma=2$ are marked.

Fig. R13 Super-resolution reconstruction of binary images by TMBH method in experiment. **a** Intensity fluctuations of reconstructed images versus increasing OAM-multiplexing channel number. **b** Comparison between TMBH and CAH methods in terms of reconstruction quality at same resolution, operating far beyond the sampling criterion limit.

Fig. R14 Experimental setup for complex amplitude modulation using two SLMs (for CAH reconstruction).

Action Taken: We have added experimental results (Fig. R12 and R13) and related discussions in the Supplementary Note 13 and 14, corresponding to the numerical results in Fig. 6 and Fig. 7 respectively.

Comment 2.7:

“6. What is the OAM purity in each channel/frame? For example, in Fig. S7, while addressing the reconstructed results of temporal multiplexing binary OAM holography, the intensity profile of OAM at different moments (T1-T50) cannot be recognized clearly.

Authors should include the discussion on the amount of OAM content at the each moment. See the following reference for more details:

a) Sroor, Hend, et al. "High-purity orbital angular momentum states from a visible metasurface laser." *Nature Photonics* 14.8 (2020): 498-503.”

Response: Thanks for the comment and suggestion. The detailed response is organized into following three parts.

1. Description of the scenario in original Fig. S7

For better understanding, we depict Fig. R15 to illustrate an example of the reconstruction results in Fig. S7. Five binary images are encoded into a single OAM multiplexed hologram through five independent OAM channels (Fig. R15a). For each of the five images, an OAM-preserved hologram is designed, and then the OAM channels are distinguished by five encoding helical phase plates ($l=2, 1, 0, -1, -2$). The image encoded in each OAM channel can be decoded by an OAM beam having a helical phase index opposite to that of the corresponding OAM channel. In the example of Fig. S7, the OAM beam with $l=2$ is used to decode an image with the content of an apple (Fig. R15b), and the decoding results of T1, T2, T49, T59 and the decoding results of TMBH (combining 50 binary holograms) are shown in Fig. S7b. On the right part of Fig. S7b, it can be seen that the pixel locations with Gaussian patterns have significantly stronger intensity distributions, and these pixel locations depict the shape of an apple, which is the basic principle of OAM beam decoding.

Fig. R15 Detailed explanation of the reconstruction example in Fig. S7. a Design flow chart of OAM multiplexed hologram. **b** Schematic diagram of the reconstruction example in Fig. S7.

2. Explain why intensity profile of OAM at different moments cannot be recognized clearly.

As the referee observes, in the reconstruction results of TMBH in Fig. S7, the OAM intensity profiles at different moments (T1-T50) cannot be clearly identified. We’re happy to explain that it has two reasons: (1) The presence of more than one OAM channels at each location results in a superimposed intensity pattern (i.e. the SPI effect). (2) The intensity profile of the OAM mode is corrupted because of the denser sampling upon the multiplexed image, which leads to the

interference of OAM modes at adjacent pixels (i.e., the API effect).

3. Measurement of OAM content in the reconstructed images

Following the referee's suggestion, we have experimentally measured the OAM content in the reconstructed images of temporal multiplexing OAM holography (TMBH), as shown in Fig. R16, as well as the OAM content in the reconstructed images at different moments of T1, T2, T49, and T50 (as shown in Fig. R17-R20 respectively). We believe this would strengthen our work.

We have carefully studied the paper suggested by the referee [Nature Photonics 14.8 (2020): 498-503], and used the parallelized topological charge analysis the paper introduced. As a result, we are able to obtain the OAM content at all positions in the reconstructed image with a single measurement. The experimental setup implementing the parallelized topological charge analysis is shown in Fig. R21.

Figure R16 shows the OAM contents of the reconstructed image after temporal multiplexing. It can be noticed that the reconstructed image contains the OAM contents of $l=0,1,2,3,4$. In particular, all the pixels containing $l=0$ form an apple pattern together (the target image), while the pixels containing $l=1,2,3,4$ form other patterns (images encoded into other OAM channels). The OAM content analysis on three representative pixels (p1, p2, p3) indicates that all OAM contents ($l=0,1,2,3,4$) are present in pixel p2, each having a near-uniform portion ($\sim 20\%$), and only the OAM content of $l=0$ is contained in pixel p1, while pixel p3 has the OAM content of $l=0,2$, as illustrated in the bottom row of Fig. R16.

In contrast, for a single moment of the reconstructed image, the content of each OAM is rather disordered, corresponding to a poorer reconstructed quality. For example, in the measurement result for $l=0$ in Fig. R17, the mode content of $l=0$ varies quite drastically, also the measured content of $l=0$ at pixel p2 is only 3%, far from the ideal value of around 20%.

Fig. R16 Topological charge decomposition of the reconstructed image after temporal multiplexing (incident OAM beam with $l=2$).

Fig. R17 Topological charge decomposition of reconstructed image at T1 moment (incident OAM beam with $l=2$).

Fig. R18 Topological charge decomposition of reconstructed image at T2 moment (incident OAM beam with $l=2$).

Fig. R19 Topological charge decomposition of reconstructed image at T49 moment (incident OAM beam with $l=2$).

Fig. R20 Topological charge decomposition of reconstructed image at T50 moment (incident OAM beam with $l=2$).

Fig. R21 Experimental setup for parallelized topological charge analysis.

Action Taken: We have presented and discussed the measured results of OAM content in the reconstructed images of TMBH and those at different moments in Supplementary Note 11.

REVIEWERS' COMMENTS

Reviewer #1 (Remarks to the Author):

Thanks for the great efforts. I am now happy to recommend it for publication.

Reviewer #2 (Remarks to the Author):

The authors have addressed my raised points and concerns very well. I am happy with the resubmitted version.